# Decarboxylative alkylation of alkenes

Triptesh Kumar Roy[1,2], Federico Maria Tamborini[1,2], Roland Petzold[1], Jianhan Fu[1], Yiben Tang[1,2] & Tobias Ritter[1✉]

Alkenes are widely used functional groups in synthetic chemistry, important for producing polymers, detergents, agrochemicals and pharmaceuticals. When treated with electrophiles, alkenes typically undergo addition, not substitution, reactions[1]. As a consequence, the intuitive retrosynthetic disconnection to form a substituted alkene from the parent alkene does not exist in the toolbox of the chemist. For example, conversion of tri-substituted into tetra-substituted alkenes, or late-stage alkylation of complex alkenes, would provide access to molecules that are currently difficult to construct. Alkene cross-metathesis can formally alkylate appropriately substituted alkenes, but diastereoselectivity and alkene–alkyl combinations are restricted to specific cases[2], and several classes of alkenes, such as internal or cyclic alkenes, cannot be readily alkylated with known methods[3]. Here we report a formal regio- and diastereoselective C−H alkylation of alkenes with carboxylic acids as alkyl source, readily available in large diversity. Key to the development is a polar decarboxylative alkylation that deviates from the current model of radical-mediated C−C bond formation from carboxylic acid derivatives, enabled by a previously unappreciated access to persistent alkylzinc intermediates from redox-active esters. A Pd-catalysed cross-coupling of the alkylzinc species with alkenyl thianthrenium salts accessed from alkenes affords the substituted alkenes in high diastereoselectivity. The transformation offers alkylation of cyclic, acyclic, terminal, internal, mono-substituted, di-substituted and tri-substituted alkenes with diverse alkyl groups.

Arenes and alkenes share the presence of $C(sp^2)$−H bonds of similar bond dissociation energy, yet their reactivity differs fundamentally. Arenes undergo electrophilic substitution because of aromatic stabilization, for example, in Friedel–Crafts alkylation. By contrast, alkenes react by electrophilic addition (Fig. 1a), which prevents analogous substitution chemistry, so there is no alkene analogue to Friedel–Crafts alkylation, and the general alkylation of alkenes remains unknown. Current strategies for substituted alkene synthesis commonly rely on alkenation reactions such as the Wittig[4], Horner–Wadsworth–Emmons (HWE)[5,6] and Julia protocols[7], which proceed from carbonyl compounds. Reductive alkylation of alkynes[8–10] offers an alternative. Alkene cross-metathesis provides a formal alkylation pathway directly from alkenes by transalkenylidination, but the types of alkene suitable for cross-metathesis with one another must be carefully selected, and the reaction is restricted to alkene pairs of appropriate reactivity[11]. Moreover, *E*-selective cross-metathesis of 1-alkenes and 1,1-di-substituted alkenes remains unknown, and several alkene classes, such as tri-substituted and cyclic alkenes, cannot be alkylated with alkene metathesis at all due to the mechanism-based alkylidene transfer that breaks the C=C double bond before a new one is formed[12–14]. An alternative approach to alkene synthesis involves cross-coupling between alkenyl nucleophiles or electrophiles and appropriate alkyl donors. Examples include alkenyl organometallic reagents that react with alkyl (pseudo)halides or redox-active esters, or conversely, alkenyl (pseudo)halides that react in cross-coupling reactions with alkyl nucleophiles or electrophiles[15–17]. However, alkenyl nucleophiles such as Grignard reagents often require multistep preparation from alkenes, show low tolerance to functional groups, and are generally not readily available in large scope[18]. Alkenyl electrophiles such as halides or triflates are, while synthetically immensely versatile[19], similarly difficult to access directly from alkenes, typically using multistep sequences, such as dibromination followed by elimination, which can form multiple constitutional isomers that reduce practicality[20]. Alkylation of alkenylthianthrenium salts is known as demonstrated in our previous work, but so far limited to a few simple, available alkylzinc reagents and restricted to terminal 1-alkenes and vinyl groups only[21–23]. Alkylation of other alkenyl sulfonium salts is confined only to styrenes[24,25]. Efforts to alkylate alkenes have so far been limited to electronically activated alkenes, such as styrenes[26] or Michael acceptors[27], and the diversity of applicable alkyl groups is narrow[28]. Heck-type alkylations typically involve the addition of alkyl halides or equivalents under palladium catalysis[26] or through photo-induced radical pathways but are generally limited to styrenes or other activated alkenes[29–31]. Alkyl carboxylic acids are excellent alkyl sources because they offer a massive, structurally diverse pool of bench-stable resources. Alkenes and alkyl carboxylic acids are among the most abundant functional groups in organic synthesis, and a general method to use them in combination for $C(sp^2)$−$C(sp^3)$ bond formation is hitherto unknown. Here we report a regio- and diastereoselective reaction to access alkylated alkenes that works across the various alkene substitution patterns and electronic environments with a large range of alkyl groups (Fig. 1b). The fundamental novelty reported herein is the recognition of a new retron for alkene alkylation, and the development of a polar decarboxylative alkylation distinct in mechanism from the established radical-mediated

[1]Max-Planck-Institut für Kohlenforschung, Mülheim an der Ruhr, Germany. [2]Institute of Organic Chemistry, RWTH Aachen University, Aachen, Germany. ✉e-mail: ritter@kofo.mpg.de

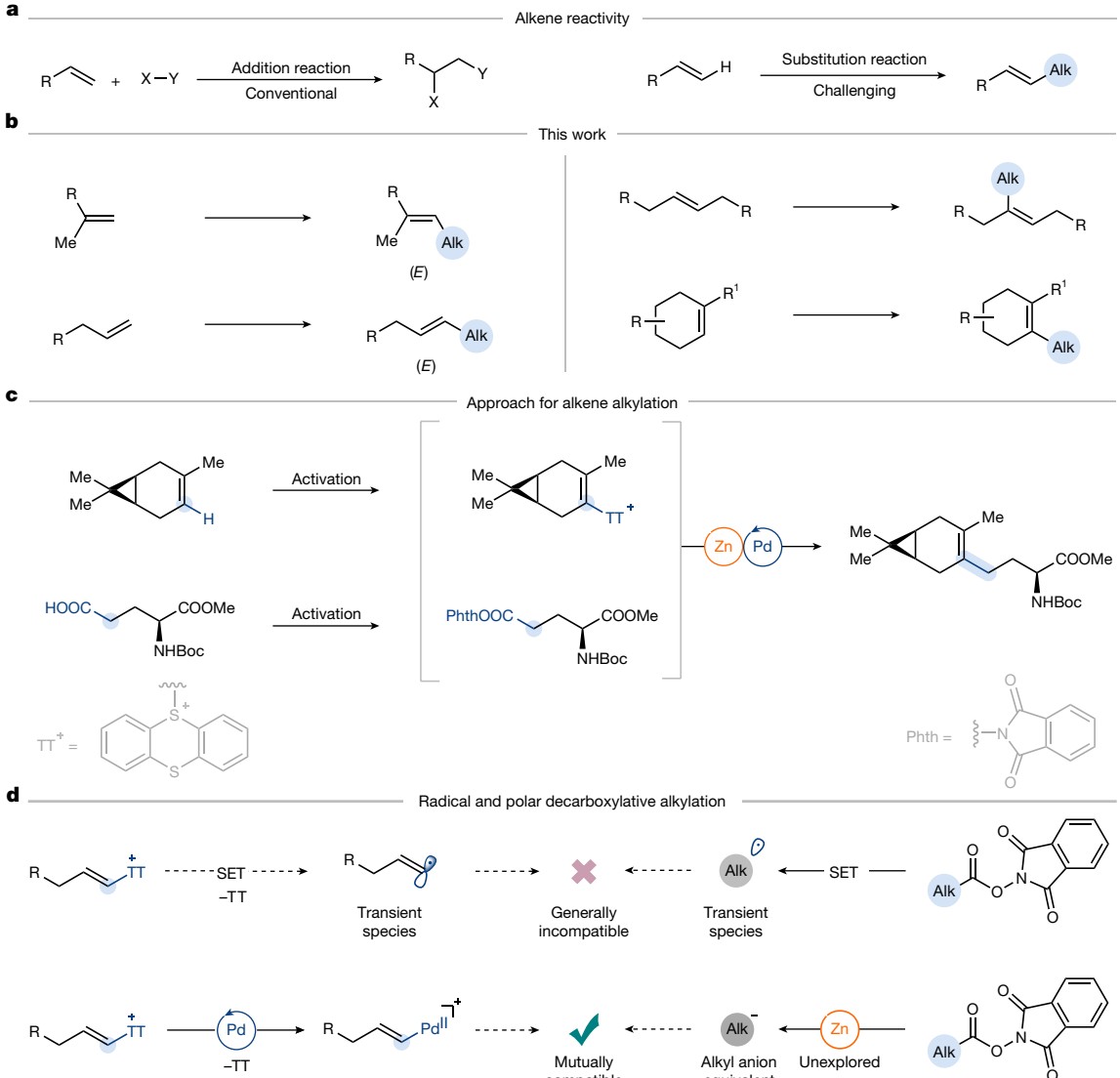

**a** ——————————————————— Alkene reactivity ———————————————————

**b** ——————————————————————— This work ———————————————————————

**c** ———————————————— Approach for alkene alkylation ————————————————

TT⁺ =

Phth =

**d** ————————— Radical and polar decarboxylative alkylation —————————

**Fig. 1 | Development of polar decarboxylative strategy for selective alkene alkylation. a**, Alkene reactivity for addition reaction and challenges with C−H alkylation of alkenes. **b**, Scope of alkene alkylation. **c**, Pd-catalysed formal decarboxylative alkylation of alkenes. **d**, Challenges with radical-mediated cross-coupling between two transient radicals formed from alkenyl thianthrenium salts and redox-active esters, and development of the polar decarboxylative C($sp^2$)−C($sp^3$) cross-coupling.

carbon−carbon bond-forming processes from activated carboxylic acids[32]. A combination of chemoselective alkylzinc formation from carboxylic acids through redox-active esters and diastereoselective alkene thianthrenation unlocks valuable chemical space and provides access to molecular architectures that are challenging to synthesize using existing methods (Fig. 1c).

Alkene C−H bonds can be substituted through thianthrenation, which proceeds by an inverse-electron-demand hetero Diels−Alder cycloaddition between the thianthrenium dication and the alkene. The resulting bicyclic dicationic intermediate undergoes base-mediated elimination through an irreversible E1cB pathway to deliver the alkenyl thianthrenium salt[20,33]. This addition−elimination sequence effectively achieves vinylic C−H substitution in a single step to provide direct access to alkenyl electrophiles from simple alkenes. Carboxylic acids, on conversion to redox-active esters, can serve as versatile alkyl group donors[34]. However, all existing methods for decarboxylative alkylation proceed through single-electron pathways, which generate alkyl radicals[34–39]. Although alkenyl thianthrenium salts can also undergo single-electron reduction ($E_p$ = −1.68 V compared with Ag/AgNO₃ in DMF), we recognized the potential difficulty to couple two non-persistent

radicals[40] (Fig. 1d); selective radical−radical cross-coupling is challenging, and, to date, such an alkyl−alkenyl bond-forming transformation has not been disclosed.

In line with the analysis of challenging cross-coupling by conventional one-electron pathways, we observed predominant protodethianthrenation of the alkenyl electrophile and hydrodecarboxylation of the redox-active ester (Supplementary Information), when we evaluated reaction conditions akin to single-electron reduction of redox-active esters with common Ni- or Fe-based catalysts[33,35,37,38] (Supplementary Table 1). These results underscore the challenge in chemoselectivity for a pathway that proceeds through C($sp^2$)−C($sp^3$) bond formation from carbon-based radicals, possibly due to competitive reduction of both electrophiles and the generation of two transient radical species. A polar, two-electron alternative through a Pd(0)/Pd(II) redox pathway with alkenylthianthrenium salts as electrophiles could improve this chemoselectivity, yet requires the conversion of redox-active esters to a carbanion synthon. A previous work[41] identified initial evidence for the conversion of alkyl redox-active esters to alkylzinc species, which has remained unexplored by the synthetic community since 2019. Based on an analysis in ref. 42, in which the reduction potential of Zn as a

**Fig. 2 | Reaction development for selective alkene alkylation. a**, Alkylzinc formation from redox-active esters. **b**, Reaction conditions for two-step alkylation of alkene. Cross-coupling reactions were conducted on 0.2 mmol scale.

function of solvent was studied, we developed the alkylzinc formation from redox-active ester to ultimately result in successful formation of persistent alkylzinc species, suitable for efficient palladium-catalysed cross-coupling with alkenylthianthrenium salts (Fig. 1d). To our knowledge, this transformation represents the first example of a polar decarboxylative C($sp^2$)–C($sp^3$) cross-coupling, enabled by formation of a stable alkylzinc intermediate that replaces the commonly used fleeting carbon-based radical as the alkylating agent. Although formation of the alkylzinc species probably proceeds through single-electron transfer (SET) from zinc on account of the electronic configuration of zinc metal, this pathway is distinct from the radical pathway in other decarboxylative bond formation, in which the carbon radical is engaged in oxidative ligation to a transition metal complex, and does not share the same chemoselectivity constraints because persistent alkylzinc species are generated instead.

Reduction of redox-active esters with zinc in solvents such as acetonitrile (MeCN) and ethyl acetate have been used for the generation of alkyl radicals[43,44]. Yet, reaction of redox-active ester **RAE-1** and Zn in DMF generated 52% of the corresponding alkylzinc species **I-1** within the first 2 h (Fig. 2a); analogous reduction of redox-active esters with Zn in solvents that are commonly used in cross-coupling chemistry, such as THF and MeCN, did not lead to alkylzinc nucleophiles synthetically useful in this transformation[45,46] (Supplementary Table 10). In-cage recombination of the incipient alkyl radical with $Zn^+$ after the first single-electron transfer and stabilization of the zinc organometallic in the polar aprotic medium may be responsible for the distinct reduction chemistry. Addition of the catalyst and alkenyl thianthrenium salt after 2 h to the reaction mixture, resulted in optimal yields for the cross-coupling (Fig. 2b). Combination of all reagents at once resulted in predominant protodethianthrenation (Supplementary Table 6). This result indicates that alkylzinc formation is slower than oxidative addition of the alkenyl thianthrenium salt to Pd(0), and initial formation of alkylzinc is essential for efficient cross-coupling. The reaction can also proceed if the redox-active ester is prepared in situ from the corresponding carboxylic acid, albeit in lower yield; for example, compound **1** was prepared in 40% yield without and in 75% yield with previous isolation of the redox-active ester (Supplementary Information). Attempts to replace Zn with alternative reductants such as Mn, Mg and organic donors such as TDAE failed to produce a cross-coupled product (Supplementary Table 5).

A large variety of different alkenes can be alkylated (Fig. 3). Because the palladium-catalysed cross-coupling from alkenyl thianthrenium

salts proceeds stereospecifically with regard to alkene geometry, and thianthrenation generally proceeds with high *E*-selectivity (>20:1 for **1**, **2**, **4**, **6**, **8** and **12**), products are produced with high stereochemical selectivity. Even 1,1-di-substituted alkenes deliver *E*-selective alkylation, for example, 10:1 for **23** and >20:1 for **20**. At present, neither compound class can be accessed through alkene cross-metathesis because a general approach for *E*-selective alkene cross-metathesis is not yet known. The Wickens group reported a method for obtaining *Z*-selective thianthrenation[22]; in our protocol, *Z*-alkenyl thianthrenium salts can be converted to *Z* alkenes under identical reaction conditions as for the cross-coupling for *E* alkenes (Supplementary Information). Electron-rich alkenes can be selectively alkylated in the presence of electron-poor alkenes, such as α,β-unsaturated ketones (**20**, **23** and **24**), because thianthrenation occurs preferentially at the most electron-rich double bond. The method is applicable to both simple feedstock alkenes and structurally complex substrates. Late-stage alkylation of terpenes such as rose oxide (**18**) and citronellol (**21**) could therefore be achieved. Aldehyde and ketone functionalities, such as those in carvone (**20**) and perillaldehyde (**23**), respectively, are well tolerated under the reaction conditions without the need for protecting groups. Terminal alkynes do not pose a chemoselectivity issue, and selective alkylation occurs exclusively at the alkene moiety without competing functionalization of the alkyne (**4**). Alkylation of tri-substituted alkenes remains synthetically challenging, with almost no precedent in the literature. The protocol here allows the selective installation of primary and secondary alkyl groups onto internal di- and tri-substituted alkenes, providing a general and predictable approach to highly substituted alkene motifs[47] (**5**, **10**, **14**, **21** and **22**). The reaction accommodates diverse alkyl partners, including strained rings, saturated heterocycles and complex derivatives such as indomethacin (**11**) and baclofen (**25**). Oxyacids, such as 2,4-dichlorophenoxyacetic acid (**17**) and menthyloxyacetic acid (**19**), can also serve as alkyl sources. The scope is limited to primary and secondary alkyl groups because tertiary redox-active esters, although reduced by zinc, do not form organozinc intermediates effectively under these conditions.

Redox-active esters ($E_p = -1.59$ V compared with Ag/AgNO$_3$ in DMF) and alkenyl thianthrenium salts ($E_p = -1.68$ V compared with Ag/AgNO$_3$ in DMF) exhibit comparable reduction potentials (Supplementary Figs. 9 and 11). However, kinetic analysis showed that zinc reduces redox-active esters significantly faster than alkenyl thianthrenium salts (Fig. 4a,b). We observed an autocatalytic effect of soluble Zn(II) for the reaction of the redox-active ester with metallic zinc and established that addition of alkylzinc accelerates chemoselective reduction, possibly through

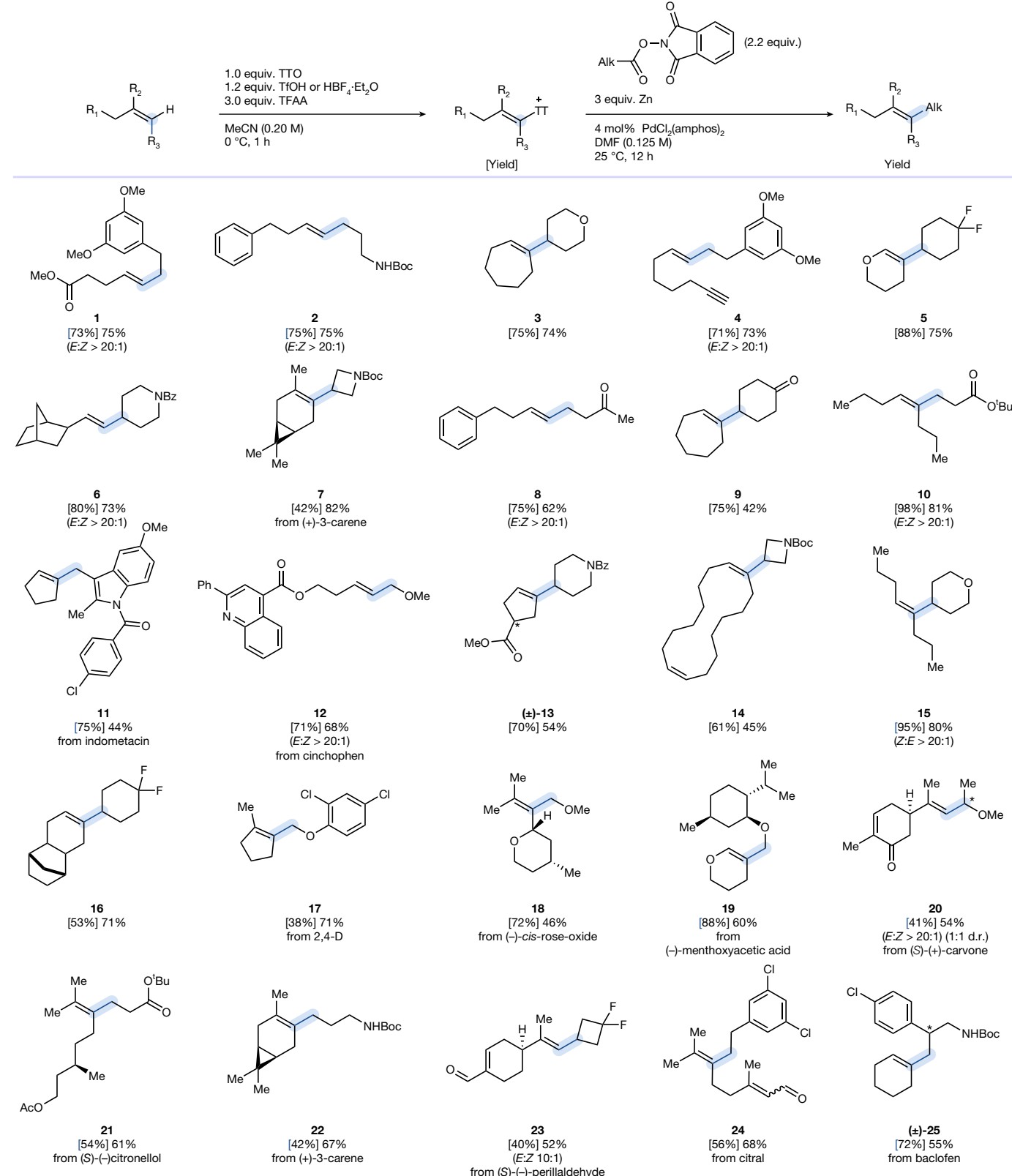

**Fig. 3 | Scope for Pd-catalysed two-step alkylation of alkenes.** Reactions for the thianthrenation of alkenes were conducted on a 0.5 mmol scale (see Supplementary Information for detailed reaction conditions). Cross-coupling reactions were conducted on a 0.2 mmol scale. General conditions for cross-coupling: alkenyl thianthrenium salt (0.20 mmol), redox-active ester (0.44 mmol), Zn dust (0.60 mmol, mesh size <60 μm), PdCl$_2$(amphos)$_2$ (4.0 mol%) and DMF (0.125 M). TTO, thianthrene-$S$-oxide; TfOH, triflic acid; TFAA, trifluoroacetic anhydride.

coordination as Lewis acid to the redox-active ester[48] (Supplementary Figs. 14–16). Analysis of the zinc-mediated reduction of **RAE-1** by [1]H and [13]C-NMR spectroscopy in DMF-$d_7$ suggests that a persistent

alkylzinc species is formed (Supplementary Figs. 3 and 4). Electrospray ionization-mass spectrometry analysis detects species consistent with alkylzinc coordinated by two DMF molecules (Supplementary Fig. 7),

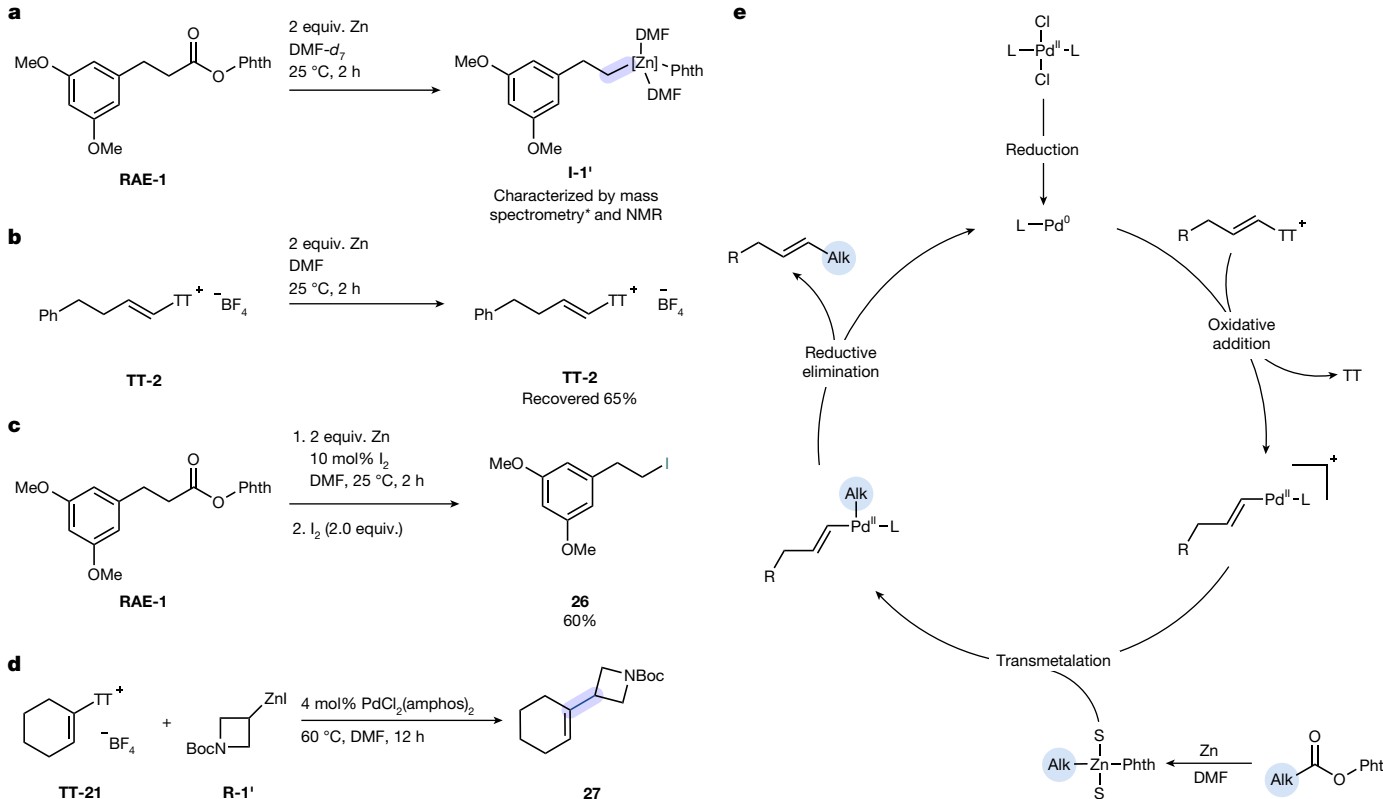

**Fig. 4 | Mechanistic studies. a**, Validation and structure of the in situ-formed alkylzinc species from redox-active esters (asterisk indicates mass spectrometry experiments were conducted in non-deuterated DMF). **b**, Control experiment for zinc-mediated reduction of alkenyl thianthrenium salt at room temperature in DMF. **c**, Iodine quenching experiment for the quantification of alkylzinc species. **d**, Cross-coupling under standard conditions with alkyl iodide and alkenyl thianthrenium salt. **e**, Plausible catalytic cycle for the polar decarboxylative cross-coupling between alkenyl thianthrenium salt and redox-active ester. S, solvent (DMF).

which suggests that solvent coordination stabilizes the organozinc intermediate. Analysis of $^1$H NMR spectra of the zinc-mediated reduction of redox-active esters revealed three major side products: alkane formation by protodecarboxylation of the redox-active ester, the corresponding carboxylic acid and a homocoupling product, all of which suggesting that decarboxylation proceeds radically, during formation of alkylzinc as the key alkylating agent (Supplementary Information). By contrast, no spectroscopic evidence for alkylzinc formation was observed when the reduction was conducted in THF-$d_8$ at either 25 °C or 60 °C (Supplementary Fig. 6). To quantify the in situ-generated alkylzinc intermediate, iodine ($I_2$) was added to the reaction mixture of redox-active ester and zinc in DMF after 2 h, which resulted in 52% yield of the alkyl iodide derived from **RAE-1** (Supplementary Information). Addition of 10 mol% $I_2$ to the alkylation reaction increased yield by 8% (Fig. 4c), consistent with in situ zinc activation. Replacement of the redox-active ester with an alkylzinc reagent directly prepared from alkyl iodide afforded cross-coupled product with 78% yield at 60 °C under otherwise identical conditions (Fig. 4d), which also supports in situ-formed alkylzinc reagent as the alkylating species for the cross-coupling reaction.

Cross-electrophile coupling is an emerging area of research that offers access to a broader pool of structurally diverse precursors than traditional electrophile–nucleophile cross-couplings[16]. In particular, decarboxylative cross-coupling between redox-active esters with other electrophiles is attractive because carboxylic acids provide abundant, bench-stable alkyl sources. Over the past decade, numerous decarboxylative strategies for C($sp^2$)–C($sp^3$) and C($sp^3$)–C($sp^3$) bond formation have transformed synthetic planning and expanded the accessible chemical space[32]. To date, decarboxylative cross-coupling approaches to C–C bond formation have been guided predominantly by radical-based retrosynthetic logic. Here, we introduce a polar decarboxylative cross-coupling pathway that generates a persistent anionic intermediate from carboxylic acids, which enables the selective coupling of two otherwise incompatible transient radicals. The development of this polar decarboxylative manifold introduces a complementary, polarity-driven disconnection strategy for C–C bond formation and establishes a new retrosynthetic logic for the alkylation of alkenes.

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

## Data availability

All data are available in the main text or the Supplementary Information.

**Acknowledgements** We thank S. Marcus, D. Margold, F. Köhler, N. Haupt and D. Kampen for mass spectrometry analysis and M. Kochius, M. Leutzsch, C. Fares, S. Tobegen and J. Jacob for NMR spectroscopy analysis. We thank P. Münstermann for her help with HPLC analysis. We thank G. Schoenn, Q. Sun, S. Müller and W. G. Whitehurst for their suggestions and discussions.

**Author contributions** T.K.R. designed the project, developed the reaction chemistry and investigated the mechanism. T.K.R., F.M.T., R.P., J.F. and Y.T. optimized and explored the substrate scope for alkene alkylation. T.K.R. and T.R. wrote the paper. T.R. directed the project.

**Funding** Open access funding provided by Max Planck Society.

**Competing interests** The authors declare no competing interests.

## Additional information

**Correspondence and requests for materials** should be addressed to Tobias Ritter.

