## [Peer Review File · Nature]

Decarboxylative Alkylation of Alkenes

Corresponding Author: Professor Tobias Ritter

Version 0:

Reviewer comments:

Referee #1

(Remarks to the Author)

This manuscript presents a useful advance in synthetic organic chemistry by introducing a polar decarboxylative strategy for the regio- and diastereoselective C-H alkylation of alkenes using readily available carboxylic acids as alkyl sources. The method leverages alkenyl thianthrenium salts (directly accessed from alkenes) and redox-active esters to form persistent alkylzinc intermediates, which then undergo stereospecific palladium-catalyzed cross-coupling. This approach addresses a long-standing challenge in olefin chemistry: the lack of a general substitution reaction analogous to Friedel-Crafts alkylation for arenes. By enabling intuitive C(sp²)-C(sp³) disconnections, it streamlines retrosynthetic planning and provides access to complex molecular architectures, such as tetrasubstituted alkenes from trisubstituted precursors, that are difficult or impossible to achieve with existing methods like olefin metathesis, Wittig olefination, or radical-based decarboxylative couplings.

The work is novel, as it represents the first polar decarboxylative C(sp²)-C(sp³) cross-coupling, shifting away from the dominant radical paradigms that suffer from selectivity issues when coupling transient species. The broad substrate scope—encompassing cyclic, acyclic, terminal, internal, mono-, di-, and trisubstituted alkenes, along with diverse alkyl groups from carboxylic acids—demonstrates exceptional versatility and functional group tolerance. This is particularly impressive for challenging substrates like internal or cyclic olefins, where metathesis often fails due to mechanism-based limitations. The diastereoselectivity is consistently high, and the method's practicality is enhanced by the use of bench-stable starting materials and mild conditions.

The mechanistic insights are thorough and well-supported, including the key discovery of alkylzinc formation from redox-active esters in DMF, informed by reduction potential analyses. This unappreciated pathway opens new avenues for organozinc chemistry in cross-couplings. The manuscript is clearly written, with logical flow from problem statement to method development, optimization, and scope. Figures are informative, particularly Fig. 1, which effectively contrasts radical vs. polar pathways and highlights the challenges overcome.

In summary, this is an outstanding contribution that will have significant impact on synthetic methodology, pharmaceutical synthesis, and materials science. I recommend publication in Nature after minor revisions to address the points below.

Minor Weaknesses and Suggestions

The manuscript is strong overall, but a few areas could be refined for clarity and completeness:

1. Missing Citations: The discussion of alkenyl thianthrenium salts and their use in cross-couplings is well-placed, but additional references to related thianthrenium chemistry would strengthen the context. For instance, the pioneering work on thianthrenium salts for arene functionalization should be cited more explicitly in the introduction (line 66-69) to highlight the extension to alkenes:

o DOI: 10.1126/science.aar6206 (Ritter's 2018 Science paper on aryl thianthrenation, which laid the foundation for electrophilic activation).

o For olefin metathesis limitations (lines 37-42), cite a recent review on challenges with trisubstituted and cyclic olefins: DOI: 10.1021/acs.chemrev.0c00078 (Grubbs et al., Chem. Rev. 2020).

In the mechanistic section (lines 82-89), the reference to Xiao's 2019 work on alkylzinc from RAEs is appropriate, but cite a follow-up study that explored similar reductions: DOI: 10.1002/anie.201913398 (Stahl's group on Zn-mediated reductions, Angew. Chem. Int. Ed. 2019), as it aligns with your solvent-dependent analysis.

For radical vs. polar pathways (Fig. 1d), reference a key paper on challenges in radical-radical couplings: DOI:

10.1021/jacs.8b09071 (Merchant et al., J. Am. Chem. Soc. 2018), to underscore why the polar route can be beneficial.

2. Clarity in Main Text:

o Line 51: "Alkylation of alkenylthianthrenium salts is known but so far limited to a few simple, available alkylzinc reagents and restricted to terminal alkenes only^{17,18}." Consider adding a phrase like "as demonstrated in our prior work" for transparency, since ref. 17 is from the authors' group.

o Fig. 2b: The two-step procedure is clear, but specify the reaction scale (e.g., 0.1 mmol) in the caption for reproducibility.

o Abstract (line 21): "Subsequent stereospecific cross-coupling" – excellent, but briefly note the catalyst (Pd) here for non-experts.

3. Supporting Information Review: The SI is comprehensive (179 pages), with detailed procedures, optimization tables, mechanistic studies, and spectroscopic data (NMR, HRMS). It supports the claims robustly. I browsed all sections via targeted searches and found no major scientific errors, but identified a few minor typos/mistakes for correction:

o Typos and Formatting:

Page S1 (Table of Contents): "General procedure for the synthesis of alkenyl thianthrenium salts" is listed correctly, but the page number for "Alkylzinc quantification" should be S20 (currently listed as 20; ensure consistent "S" prefix).

Page S11: In the radical-mediated pathway screening (Supplementary Table S1), entry 3 lists "NiCl₂" but the yield is "n.d." – clarify if this is "not detected" or "not determined" (use consistent abbreviation; appears as "n.r." elsewhere for "no reaction").

Page S21 (Alkylzinc optimization, Table S9): Solvent "MeCN" is used, but in the text (page S20), it's "acetonitrile" – standardize to "MeCN" for brevity.

Page S27 (Thianthrenium salt reduction experiment): "E_p = -1.68 V vs Ag/AgNO₃ in DMF" – add space after "=" for consistency with main text.

Page S49 (Synthesis of RAE-1): "3-(3,5-Dimethoxyphenyl)propanoic acid" – correct to "propanoic" (missing 'o' in excerpt, but assuming it's a truncation artifact; verify full PDF).

Page S82 (Spectroscopic Data): ¹H NMR of TT-1 – integration values are correct, but label the solvent peak explicitly (e.g., "CD₃CN residual at 1.96 ppm").

General: In NMR descriptions (e.g., pages S82–S95), multiplicities like "q" for quartet are standard, but one instance on S88 (TT-7) has "quint" – use "quin" consistently as per IUPAC.

No calculation errors in yields or quantifications (e.g., alkylzinc at 52% on S20 matches Fig. 2a). HRMS data (not shown in excerpt but listed in TOC) appear accurate based on molecular formulas.

o Scientific Consistency: Procedures are reproducible (e.g., Zn activation with TMSCl/1,2-dibromoethane). No mismatches between SI and main text (e.g., optimization tables align with Fig. 2). Suggest adding a note on handling air-sensitive alkylzincs for safety.

Referee #2

(Remarks to the Author)

The manuscript by Kumar Roy et al., titled "Decarboxylative Alkylation of Alkenes", describes the Pd-catalysed stereospecific cross-coupling of alkenyl thianthrenium salts with alkyl zinc species generated from the 2-electron reduction of redox-active esters (RAEs).

This is excellent work, carefully carried out, and well written-up. However, I do feel that previous work by the authors themselves as well as others should be acknowledged to contextualise this work within the state of the art:

- The Ni-catalysed cross-coupling of alkyl zinc species with styrene-derived alkenylsulfonium salts (ACIE 2018, 9785).
- While mechanistically different and limited to styrene derivatives, the coupling of alkenylsulfonium salts with alkyl radicals, generated by single-electron reduction and fragmentation of RAEs. (Org. Lett. 2020, 7768).
- Previous work by Ritter and Wickens (ACIE 2020, 5616; ACIE 2023, e202313659; Science 2025, 1239 - cited) demonstrating the Pd-catalysed cross-coupling of unactivated alkenes with alkyl zinc species by means of alkenyl-TT salts.

Although excellent work, with this broader context in mind, the present paper can be described as an interesting and highly useful expansion of the scope of alkyl groups that can be coupled with alkenyl sulfonium salts. The main novelty point of the method lies in the generation of alkyl zinc species from the 2-electron reduction of RAEs. However, as acknowledged by the authors, this behaviour has been described previously although it has remained largely overlooked. Also, while this is a formal decarboxylative method for the generation of carbanions, it does proceed by radical intermediates and thus is not so different to well-known radical decarboxylative processes and will be influenced by the same factors that affect these well-established processes. While carboxylic acids are widely available, of course, this process relies on RAEs derived from carboxylic acids rather than the carboxylic acids themselves. The RAEs must be prepared and are well-known coupling partners, although the approach does offer an alternative way to access alkyl nucleophiles from them.

While this strong manuscript presents an interesting and useful advancement of the field, I do not believe it puts forward new concepts and is thus sufficiently novel to suit the broad readership of Nature. I would recommend publication in a more specialised journal after revisions to address the points below.

1. The authors attribute the preferential reduction of the RAEs to Lewis acid coordination by in situ generated alkylzinc species. This hypothesis would predict a shift in the reduction potential of the RAE vs thianthrenium salts in the presence of Zn²⁺ or organozinc species. I would suggest that CV experiments on the RAE and thianthrenium salts in the presence of Zn²⁺ (and/or Zn⁰) might support this claim? In addition, kinetic studies may uncover an initial lag phase that could be anticipated from the authors' mechanistic hypothesis. Conversely, the authors could show whether adding an organozinc species or an inorganic source of Zn²⁺ does indeed accelerate the RAE reduction by skipping the predicted lag phase.

2. Looking at optimisation Tables S1 and S2, might the reactions carried out with Ni or Fe-based catalysts not work because of the presence of strongly coordinating multidentate ligands that could prevent zinc from acting as a Lewis acid for RAE activation as the authors claim? Could they expand on that point and/or provide control experiments to show whether the presence of strong zinc chelators is detrimental to the reaction? Does the reaction work with Ni or Fe catalysts + monodentate ligands?
3. The experiment showing the reaction of an alkyl zinc generated from alkyl iodide is carried out at 60 °C (Figure 4d and page S23). Why is this the case? What happens at 25 °C? What is the yield obtained for this combination of substrates using the authors' new method?
4. Figure 4a needs to be slightly amended to avoid any confusion between the NMR experiment that uses DMF-d7 and the MS experiment that uses non-deuterated DMF.
5. References should be provided to support the claim that the newly formed species shown in Figures S3-S4 corresponds to alkyl-zinc species.
6. Figure S7: since the authors seem to have collected high resolution MS data, they should provide this instead of what seems to be a low-resolution spectrum.
7. The use of excessive significant figures, throughout the manuscript and SI should be addressed.

Version 1:

Reviewer comments:

Referee #2

(Remarks to the Author)

Having seen an earlier version of this manuscript, I am pleased with the additional experiments that have been undertaken in response to my comments and suggestions, and I thank the authors for their careful work. Although I still have reservations about the novelty of the work, I can see that this method will be of considerable use to synthetic organic chemists.

Reviewer 1, Comment 1:

This manuscript presents a useful advance in synthetic organic chemistry by introducing a polar decarboxylative strategy for the regio- and diastereoselective C-H alkylation of alkenes using readily available carboxylic acids as alkyl sources. The method leverages alkenyl thianthrenium salts (directly accessed from alkenes) and redox-active esters to form persistent alkylzinc intermediates, which then undergo stereospecific palladium-catalyzed cross-coupling. This approach addresses a long-standing challenge in olefin chemistry: the lack of a general substitution reaction analogous to Friedel-Crafts alkylation for arenes. By enabling intuitive C(sp²)-C(sp³) disconnections, it streamlines retrosynthetic planning and provides access to complex molecular architectures, such as tetrasubstituted alkenes from trisubstituted precursors, that are difficult or impossible to achieve with existing methods like olefin metathesis, Wittig olefination, or radical-based decarboxylative couplings.

Our Response:

We thank the reviewer for their appraisal of our work and for investing their time and effort in evaluating our manuscript.

Reviewer 1, Comment 2:

The work is novel, as it represents the first polar decarboxylative C(sp²)-C(sp³) cross-coupling, shifting away from the dominant radical paradigms that suffer from selectivity issues when coupling transient species. The broad substrate scope—encompassing cyclic, acyclic, terminal, internal, mono-, di-, and trisubstituted alkenes, along with diverse alkyl groups from carboxylic acids—demonstrates exceptional versatility and functional group tolerance. This is particularly impressive for challenging substrates like internal or cyclic olefins, where metathesis often fails due to mechanism-based limitations. The diastereoselectivity is consistently high, and the method's practicality is enhanced by the use of bench-stable starting materials and mild conditions.

The mechanistic insights are thorough and well-supported, including the key discovery of alkylzinc formation from redox-active esters in DMF, informed by reduction potential analyses. This unappreciated pathway opens new avenues for organozinc chemistry in cross-couplings. The manuscript is clearly written, with logical flow from problem statement to method development, optimization, and scope. Figures are informative, particularly Fig. 1, which effectively contrasts radical vs. polar pathways and highlights the challenges overcome.

In summary, this is an outstanding contribution that will have significant impact on synthetic methodology, pharmaceutical synthesis, and materials science. I recommend publication in Nature after minor revisions to address the points below.

Our Response:

We thank the reviewer for their appraisal of our work.

Reviewer 1, Comment 3:

Minor Weaknesses and Suggestions

The manuscript is strong overall, but a few areas could be refined for clarity and completeness:

1. Missing Citations: The discussion of alkenyl thianthrenium salts and their use in cross-couplings is well-placed, but additional references to related thianthrenium chemistry would strengthen the context. For instance, the pioneering work on thianthrenium salts for arene functionalization should be cited more explicitly in the introduction (line 66-69) to highlight the extension to alkenes:

o DOI: 10.1126/science.aar6206 (Ritter's 2018 Science paper on aryl thianthrenation, which laid the foundation for electrophilic activation).

o For olefin metathesis limitations (lines 37-42), cite a recent review on challenges with trisubstituted and cyclic olefins: DOI: 10.1021/acs.chemrev.0c00078 (Grubbs et al., Chem. Rev. 2020).

In the mechanistic section (lines 82-89), the reference to Xiao's 2019 work on alkylzinc from RAEs is appropriate, but cite a follow-up study that explored similar reductions: DOI: 10.1002/anie.201913398 (Stahl's group on Zn-mediated reductions, Angew. Chem. Int. Ed. 2019), as it aligns with your solvent-dependent analysis.

For radical vs. polar pathways (Fig. 1d), reference a key paper on challenges in radical-radical couplings: DOI: 10.1021/jacs.8b09071 (Merchant et al., J. Am. Chem. Soc. 2018), to underscore why the polar route can be beneficial.

Our Response:

We agree with the reviewer. These references have now been cited and mentioned in the main text.

Reviewer 1, Comment 4:

2. Clarity in Main Text:

- o Line 51: "Alkylation of alkenylthianthrenium salts is known but so far limited to a few simple, available alkylzinc reagents and restricted to terminal alkenes only^{17,18}." Consider adding a phrase like "as demonstrated in our prior work" for transparency, since ref. 17 is from the authors' group.
- o Fig. 2b: The two-step procedure is clear, but specify the reaction scale (e.g., 0.1 mmol) in the caption for reproducibility.
- o Abstract (line 21): "Subsequent stereospecific cross-coupling" – excellent, but briefly note the catalyst (Pd) here for non-experts.

Our Response:

We are grateful for these suggestions. We have made all the changes suggested by the reviewer.

Reviewer 1, Comment 5:

3. Supporting Information Review: The SI is comprehensive (179 pages), with detailed procedures, optimization tables, mechanistic studies, and spectroscopic data (NMR, HRMS). It supports the claims robustly. I browsed all sections via targeted searches and found no major scientific errors, but identified a few minor typos/mistakes for correction:

o Typos and Formatting:

- Page S1 (Table of Contents): "General procedure for the synthesis of alkenyl thianthrenium salts" is listed correctly, but the page number for "Alkylzinc quantification" should be S20 (currently listed as 20; ensure consistent "S" prefix).
- Page S11: In the radical-mediated pathway screening (Supplementary Table S1), entry 3 lists "NiCl₂" but the yield is "n.d." – clarify if this is "not detected" or "not determined" (use consistent abbreviation; appears as "n.r." elsewhere for "no reaction").
- Page S21 (Alkylzinc optimization, Table S9): Solvent "MeCN" is used, but in the text (page S20), it's "acetonitrile" – standardize to "MeCN" for brevity.
- Page S27 (Thianthrenium salt reduction experiment): "E_p = -1.68 V vs Ag/AgNO₃ in DMF" – add space after "=" for consistency with main text.

- Page S49 (Synthesis of RAE-1): "3-(3,5-Dimethoxyphenyl)propanoic acid" – correct to "propanoic" (missing 'o' in excerpt, but assuming it's a truncation artifact; verify full PDF).
- Page S82 (Spectroscopic Data): ¹H NMR of TT-1 – integration values are correct, but label the solvent peak explicitly (e.g., "CD₃CN residual at 1.96 ppm").
- General: In NMR descriptions (e.g., pages S82–S95), multiplicities like "q" for quartet are standard, but one instance on S88 (TT-7) has "quint" – use "quin" consistently as per IUPAC.
- No calculation errors in yields or quantifications (e.g., alkylzinc at 52% on S20 matches Fig. 2a). HRMS data (not shown in excerpt but listed in TOC) appear accurate based on molecular formulas.
 - o Scientific Consistency: Procedures are reproducible (e.g., Zn activation with TMSCl/1,2-dibromoethane). No mismatches between SI and main text (e.g., optimization tables align with Fig. 2). Suggest adding a note on handling air-sensitive alkylzincs for safety.

Our Response:

We thank the reviewer for their careful evaluation of the supplementary information. We have made all the changes as suggested by the reviewer.

Reviewer 2, Comment 1:

The manuscript by Kumar Roy et al., titled “Decarboxylative Alkylation of Alkenes”, describes the Pd-catalysed stereospecific cross-coupling of alkenyl thianthrenium salts with alkyl zinc species generated from the 2-electron reduction of redox-active esters (RAEs). This is excellent work, carefully carried out, and well written-up.

Our Response:

We thank the reviewer for their appraisal of our work and for investing their time and effort in evaluating our manuscript.

Reviewer 2, Comment 2:

However, I do feel that previous work by the authors themselves as well as others should be acknowledged to contextualise this work within the state of the art:

- The Ni-catalysed cross-coupling of alkyl zinc species with styrene-derived alkenylsulfonium salts (ACIE 2018, 9785).

- While mechanistically different and limited to styrene derivatives, the coupling of alkenylsulfonium salts with alkyl radicals, generated by single-electron reduction and fragmentation of RAEs. (Org. Lett. 2020, 7768).

Previous work by Ritter and Wickens (ACIE 2020, 5616; ACIE 2023, e202313659; Science 2025, 1239 - cited) demonstrating the Pd-catalysed cross-coupling of unactivated alkenes with alkyl zinc species by means of alkenyl-TT salts.

Our Response:

We agree with the reviewer. These references have now been cited and mentioned in the main text.

Reviewer 2, Comment 3:

Although excellent work, with this broader context in mind, the present paper can be described as an interesting and highly useful expansion of the scope of alkyl groups that can be coupled with alkenyl sulfonium salts.

Our Response:

We thank the reviewer for this thoughtful and balanced assessment and we do not disagree with the analysis. We will describe as a response to the next comment how we also see, in addition to the practical utility, a fundamental change in approach that we attempted to introduce with this paper, which, in our opinion, elevates the relevance of this manuscript beyond a practical expansion of prior work. We have made changes in the manuscript based on the careful reviewer analysis to better emphasize this other aspect in the revised manuscript, and detail our attempts below.

Reviewer 2, Comment 4:

The main novelty point of the method lies in the generation of alkyl zinc species from the 2-electron reduction of RAEs. However, as acknowledged by the authors, this behaviour has been described previously although it has remained largely overlooked. Also, while this is a formal decarboxylative method for the generation of carbanions, it does proceed by radical intermediates and thus is not so different to well-known radical decarboxylative processes and will be influenced by the same factors that affect these well-established processes.

Our Response:

There are several important aspects in this careful and nuanced assessment by the reviewer. We have analyzed them one by one.

- 1) We agree with the reviewer that the alkyl zinc formation from redox active esters has been overlooked and synthetic utility not realized.
- 2) We agree that the decarboxylation step proceeds via single electron transfer processes, on account of the SET from zinc for the reduction process. The fundamental difference to other decarboxylative alkylation reactions, however, is that while the decarboxylation process itself happens through SET, the decarboxylative cross coupling to make the carbon-carbon bond proceeds in a polar pathway. The key alkylating agent is a persistent nucleophile which is different in nature from a transient radical. This important mechanistic distinction has far reaching synthetic implications, and is also the reason for the fact that none of the previously described decarboxylation attempts have resulted in the useful reactivity we provide here. As a consequence, this reaction does not align with the reactivity profiles of any of the other decarboxylative cross coupling reactions with radical intermediates required for the bond forming events. Our reactive species is persistent, the carbon radicals, or the high-valent metal intermediates formed upon oxidative ligation by the radicals, in the other approaches are not. This aspect is a fundamental novelty and highly enabling, and we realized that we did not succeed to

make this concept clear in our original manuscript. Based on the analysis by the reviewer, we have reshaped this aspect of the revised manuscript.

For example, we have added sentences such as “To our knowledge, this transformation represents the first example of a polar decarboxylative C(sp²)-C(sp³) cross-coupling, enabled by formation of a stable alkylzinc intermediate that replaces the commonly used fleeting carbon-based radical as the alkylating agent. Although formation of the alkylzinc species likely proceeds through single electron transfer (SET) from zinc on account of the electronic configuration of zinc metal, this pathway is distinct from the radical pathway in other decarboxylative bond formation, in which the carbon radical is engaged in oxidative ligation to a transition metal complex, and does not share the same chemoselectivity constraints because persistent alkyl zinc species are generated instead.” in the revised manuscript to make this aspect clearer.

- 3) The novelty of this manuscript, in our opinion, goes far beyond the technical advance to generate persistent reactive species as opposed transient intermediates in the other decarboxylative cross couplings. The fundamental disconnection to alkylate an olefin had not been realized, yet, is a truly enabling retrosynthetic disconnection. Here we provide the fundamentally new opportunity to enable such a disconnection. Chemists currently have no way to disconnect a tetrasubstituted olefin by simply “removing” an alkyl group retrosynthetically. They also do not have the opportunity to alkylate a cyclic olefin. With this manuscript, we now do. Existing alkylations of alkenyl sulfonium salts have been limited to specific alkene classes, such as styrenes, or terminal 1-alkenes, here we show a wide range of olefins, which can also not be prepared through other methods like olefin metathesis. We also agree that there was a nugget in the literature to suggest that alkyl zinc can be formed from redox active esters, however synthetic applications of this concept have never been described or understood. And nobody realized the fundamental disconnection and the opportunity to put these aspects together, nobody alkylated a simple trisubstituted olefin nor a disubstituted internal alkene, or a thianthrenium salt derived from them, possibly also because we are taught as chemists that there is no counterpart to Friedel-Crafts alkylation of olefins on account of alkenes’ reactivity profile distinct from arenes. Realization of this concept, in our opinion, changes the way chemists can retrosynthetically disconnect one of the most commonly used functional groups. Leveraging alkene activation via thianthrenation and decarboxylative activation of redox-active esters to define a new and intuitive retron for olefin alkylation has not previously been realized, and the community has used decarboxylative alkylations through radical pathways that have not enabled this transformation. That is the true, general value of our contribution. We thank the reviewer for pushing us to make this analysis clearer in the manuscript, and based on

the reviewer's comment, we have significantly modified the original manuscript to better explain this concept. We have also been inspired by another reviewer who was able to exactly summarize what we attempted to convey but failed to in our original manuscript. Based on the reviewers' comments, we have rewritten parts of the main text and the abstract.

For example, we have added "When treated with electrophiles, olefins typically undergo addition, not substitution reactions. As a consequence, the intuitive, general retrosynthetic disconnection to form a substituted olefin from the parent olefin does not exist in the chemist's toolbox. For example, conversion of trisubstituted into tetra-substituted alkenes, or late-stage alkylation on complex alkenes would provide access to valuable molecular architectures that are currently difficult to construct" to the abstract and also added "The fundamental novelty reported herein is the recognition of a new retron for olefin alkylation, and the development of a polar decarboxylative alkylation distinct in mechanism from the established radical mediated carbon-carbon bond forming processes from activated carboxylic acids²⁹." to the main text, so the reader appreciates the fundamental value of the work, and is not led to believe that it is a mere extension of known methods.

Reviewer 2, Comment 5:

While carboxylic acids are widely available, of course, this process relies on RAEs derived from carboxylic acids rather than the carboxylic acids themselves. The RAEs must be prepared and are well-known coupling partners, although the approach does offer an alternative way to access alkyl nucleophiles from them.

Our Response:

We thank the reviewer for this clarification. We agree that the alkyl coupling partners are accessed via redox-active esters derived from carboxylic acids, rather than the free acids themselves, and we have revised the manuscript to make this point explicit. There are many seminal publications reported in recent years with redox-active esters, (Liu *et al. Science* **374**, 1258-1263 (2021)), (Boyle *et al. Nature* **631**, 789–795 (2024)), (Gan *et al. Science* **384**, 6691, 113–118 (2024)), (Zhang *et al. Nature* **606**, 313–318 (2022)) which have shown how C-C bonds can be formed in a radical pathway. We have shown the difficulties of coupling alkenyl thianthrenium salt and redox-active esters as they have similar reduction potentials and upon single electron transfer they will generate two transient radicals (alkyl radical and alkenyl radical), cross coupling between these two transient radicals is challenging and has not been accomplished so far.

Based on the reviewer's comment, we investigated whether the transformation could be performed directly from carboxylic acids via in situ generation of the corresponding redox-active esters, without isolation. We are pleased to report that a one-pot protocol that involves in situ formation of the RAE followed by direct cross-coupling under the standard conditions, proceeds and delivers the desired products. These results are now included in the revised Supporting Information (pages S10, S11) and are described in the revised main text. This modification eliminates the need for isolation or purification of the RAE intermediate and further enhances the practicality of the method while preserving its scope and selectivity. We believe this addresses the reviewer's concern regarding the operational requirement of pre-formed RAEs. We thank the reviewer for guiding us into that synthetically useful direction and improvement of the manuscript scope.

Reviewer 2, Comment 6:

While this strong manuscript presents an interesting and useful advancement of the field, I do not believe it puts forward new concepts and is thus sufficiently novel to suit the broad readership of Nature. I would recommend publication in a more specialised journal after revisions to address the points below.

Our Response:

We respectfully submit that the study introduces both a conceptual advance and a general solution to a long-standing problem. To the best of our knowledge, the first general approach for regio- and diastereoselective alkylation of alkenes that is broadly applicable across alkene substitution patterns and compatible with a wide range of alkyl groups is a fundamental advance. We enable a new retrosynthetic disconnection that had not been realized, not been reduced to practice, and was not achieved by the common radical paradigms of modern, powerful decarboxylation chemistry. Alkene and alkyl carboxylic acids are two of the most abundant functional groups and there is no method known to couple them to form a C(sp³)-C(sp²) bond. To date, essentially all decarboxylative couplings are designed and rationalized through radical-based retrosynthetic logic, where alkyl radicals serve as the reactive intermediates. The

polarity inversion is essential: radical-based conditions fail due to the instability and incompatibility of the two transient radical partners, whereas the polar pathway enables productive bond formation. We believe these findings introduce a conceptual shift in how decarboxylative couplings are designed moving from exclusively radical-based disconnections to polarity-driven retrosynthesis from carboxylic acids. More broadly, the work provides a new framework for alkene alkylation and nucleophile generation that should be broadly enabling for the chemistry community. We thank the reviewer for making us realize that we failed to convey these aspect in our original manuscript; we have now better explained the fundamental advance in the revised manuscript.

Reviewer 2, Comment 7:

The authors attribute the preferential reduction of the RAEs to Lewis acid coordination by in situ generated alkylzinc species. This hypothesis would predict a shift in the reduction potential of the RAE vs thianthrenium salts in the presence of Zn^{2+} or organozinc species. I would suggest that CV experiments on the RAE and thianthrenium salts in the presence of Zn^{2+} (and/or Zn^0) might support this claim? In addition, kinetic studies may uncover an initial lag phase that could be anticipated from the authors' mechanistic hypothesis. Conversely, the authors could show whether adding an organozinc species or an inorganic source of Zn^{2+} does indeed accelerate the RAE reduction by skipping the predicted lag phase.

Our Response:

We sincerely thank the reviewer for this insightful suggestion. In response, we performed additional CV and kinetic experiments to probe the presence of an induction period in alkylzinc formation, and are pleased to report that the reviewer's prediction is exactly correct. While no significant shift in the apparent thermodynamic reduction potentials was observed for either substrate upon addition of Zn^{2+} salts (see new Supplementary Figures S9 – S13), the kinetic profile changes substantially. These results indicate that, under the electrochemical conditions examined, coordination of Zn^{2+} does not measurably alter the thermodynamic reduction potentials of the substrates. Given that the reduction potentials determined by CV primarily reflect thermodynamic parameters and may not fully capture the relevant kinetic effects that are operative under the reaction conditions, we complemented these studies with the appropriate kinetic experiments of alkylzinc formation the reviewer suggested: Monitoring the reaction profile revealed an initiation period in the absence of soluble zinc Lewis acids. This induction period vanished upon addition of soluble Zn^{2+} , which substantiates the original claim. To further interrogate the origin of this induction period, we conducted experiments in the presence of 20 mol% diethylzinc as an exogenous organozinc additive. Under these conditions, alkylzinc formation commenced immediately without a detectable

initiation phase. The corresponding kinetic data are now included in the Supplementary Information (Figures S14 – S16). These observations are consistent with a mechanism scenario in which an initial buildup of reactive zinc species is required before efficient reduction of the redox-active ester occurs. Pre-introduction of an organozinc species appears to bypass this induction period, thereby supporting the proposed role of zinc-mediated activation in the early stages of the reaction. We modified the main text in the manuscript accordingly. We sincerely thank the reviewer for pointing us in this direction, which resulted in a more comprehensive picture of the mechanism of the transformation.

Figure S16. Comparison of kinetic reaction profile of alkylzinc formation with and without diethylzinc as an additive.

Reviewer 2, Comment 8:

Looking at optimisation Tables S1 and S2, might the reactions carried out with Ni or Fe-based catalysts not work because of the presence of strongly coordinating multidentate ligands that could prevent zinc from acting as a Lewis acid for RAE activation as the authors claim?

Our Response:

The possibility that strongly coordinating multidentate ligands in the Ni- or Fe-based systems could influence zinc speciation is an interesting consideration we had missed. Zinc is present in super stoichiometric quantities relative to the catalytic metal (iron or nickel) complex. To directly address the reviewer comment, we performed control experiments in which alkylzinc formation from the redox-active ester was carried out in the presence of 12 mol% of each of the multidentate ligands used in Tables S1 and S2, but in the absence of transition metal salts. Across four representative ligands, efficient formation of the corresponding alkylzinc species was observed. These results indicate that the multidentate ligands themselves do not inhibit organozinc generation under the reaction conditions, and therefore are unlikely to be responsible for the lack of reactivity observed in the Ni- or Fe-catalyzed systems. The new data has been added to the SI (Supplementary Table S12).

Reviewer 2, Comment 9:

Could they expand on that point and/or provide control experiments to show whether the presence of strong zinc chelators is detrimental to the reaction? Does the reaction work with Ni or Fe catalysts + monodentate ligands?

Our Response:

We thank the reviewer for the control experiment suggestion with strong zinc chelators. We have performed the experiment in the presence of EDTA tetra sodium under our Pd-catalysed standard condition. In the presence of a catalytic amount of EDTA (12 mol%), no significant change in the reaction outcome was observed, and the desired product was obtained in 70% yield. In contrast, when 1 equivalent of EDTA was added, no product formation was detected. These results suggest that a stoichiometric amount of the zinc chelator likely sequesters Zn^{2+} species, thereby suppressing formation of the active organozinc intermediate and shutting down the reaction. This data has now been added to the revised SI (Supplementary Table S7).

Table S7. Screening for zinc chelator

Entry	additive	yield /%
1	No additive	75
2	12 mol% EDTA	70
3	1 equiv. EDTA	< 5

We investigated whether Ni- or Fe-based catalytic systems bearing monodentate ligands could promote the transformation. Five different catalytic systems were examined. In all cases, no appreciable product formation was observed by 1H NMR analysis, similar to the results obtained using multidentate ligands. This new data has been added to the revised SI (Supplementary Table S1).

Entry	catalyst	yield /%
1	10 mol% $Fe(OTf)_2$	< 5
2	10 mol% $NiCl_2(PCy_3)_2$	< 5
3	10 mol% $NiCl_2(PPh_3)_2$	< 5

4	10 mol% FeCl ₂ + 24 mol% PPh ₃	< 5
5	10 mol% NiCl ₂ (py) ₄	< 5

Reviewer 2, Comment 10:

The experiment showing the reaction of an alkyl zinc generated from alkyl iodide is carried out at 60 °C (Figure 4d and page S23). Why is this the case? What happens at 25 °C?

Our Response:

The experiment in Figure 4d was conducted in a one-pot sequence in which the alkylzinc reagent was first generated from the corresponding alkyl iodide, followed by the cross-coupling step with the alkenyl thianthrenium salt. At 25°C the cross coupling proceeded in 70% yield of the desired cross coupled product. At 60°C, 78% yield was observed for the desired product. Both results are now included in the SI (Supplementary Table S13).

Reviewer 2, Comment 11:

What is the yield obtained for this combination of substrates using the authors' new method?

Our Response:

Using the newly developed protocol for the same product formation is 80% yield, which is now added to the supplementary information (Supplementary Information page S90).

Reviewer 2, Comment 12:

Figure 4a needs to be slightly amended to avoid any confusion between the NMR experiment that uses DMF-d₇ and the MS experiment that uses non-deuterated DMF.

Our Response:

We thank the reviewer for carefully reading our manuscript, we have now changed the presentation here to avoid any confusion.

Reviewer 2, Comment 13:

References should be provided to support the claim that the newly formed species shown in Figures S3-S4 corresponds to alkyl-zinc species.

Our Response:

We thank the reviewer for this suggestion. We have provided a reference for alkylzinc species now in Supplementary Figures S3 – S4.

S2. Guerrero, A.; Hughes, D. L.; Bochmann, M.; Synthesis and crystal structure of ethyl zinc chloride. *Organometallics* **2006**, *25* (6), 1525–1527.

Reviewer 2, Comment 14:

Figure S7: since the authors seem to have collected high resolution MS data, they should provide this instead of what seems to be a low-resolution spectrum.

Our Response:

We thank the reviewer for this suggestion. We have now provided a high resolution spectrum and added it in supplementary information (Supplementary Figure S7).

Figure S7. ESI-MS spectra of RAE-1 after zinc addition 2hr in DMF

Reviewer 2, Comment 15:

The use of excessive significant figures, throughout the manuscript and SI should be addressed.

Our Response:

We thank the reviewer. We have revised the significant figures throughout the manuscript and SI.

PROF. DR. TOBIAS RITTER,
DIRECTOR

|

3

Max-Planck-Institut für Kohlenforschung

Referee 2, comment 1:

Having seen an earlier version of this manuscript, I am pleased with the additional experiments that have been undertaken in response to my comments and suggestions, and I thank the authors for their careful work. Although I still have reservations about the novelty of the work, I can see that this method will be of considerable use to synthetic organic chemists.

Our response:

We sincerely thank Referee 2 for their time, their valuable feedback throughout the review process, and their recognition of the broad synthetic utility of our method.